# Soft gamma rays from low accreting supermassive black holes and connection to energetic neutrinos

Shigeo S. Kimura [1,2✉], Kohta Murase[3,4,5,6] & Péter Mészáros [3,4,5]

The Universe is filled with a diffuse background of MeV gamma-rays and PeV neutrinos, whose origins are unknown. Here, we propose a scenario that can account for both backgrounds simultaneously. Low-luminosity active galactic nuclei have hot accretion flows where thermal electrons naturally emit soft gamma rays via Comptonization of their synchrotron photons. Protons there can be accelerated via turbulence or reconnection, producing high-energy neutrinos via hadronic interactions. We demonstrate that our model can reproduce the gamma-ray and neutrino data. Combined with a contribution by hot coronae in luminous active galactic nuclei, these accretion flows can explain the keV – MeV photon and TeV – PeV neutrino backgrounds. This scenario can account for the MeV background without non-thermal electrons, suggesting a higher transition energy from the thermal to nonthermal Universe than expected. Our model is consistent with X-ray data of nearby objects, and testable by future MeV gamma-ray and high-energy neutrino detectors.

[1] Frontier Research Institute for Interdisciplinary Sciences, Tohoku University, Sendai, Japan. [2] Astronomical Institute, Tohoku University, Sendai, Japan. [3] Department of Physics, The Pennsylvania State University, University Park, PA, USA. [4] Department of Astronomy & Astrophysics, The Pennsylvania State University, University Park, PA, USA. [5] Center for Multimessenger Astrophysics, Institute for Gravitation and the Cosmos, The Pennsylvania State University, University Park, PA, USA. [6] Center for Gravitational Physics, Yukawa Institute for Theoretical Physics, Kyoto, Japan. ✉email: shigeo@astr.tohoku.ac.jp

The Universe is filled with high-energy radiation including X-rays[1], MeV–TeV gamma-rays[2,3], and TeV–PeV neutrinos[4]. It is widely believed that the cosmic X-ray background (CXB) is predominantly produced by radio-quiet active galactic nuclei (AGN) and star-forming galaxies[5,6], and that radio-loud AGN including blazars and star-forming galaxies provide the bulk of the GeV–TeV gamma-ray backgrounds[7,8]. However, the origins of the soft gamma-ray and high-energy neutrino backgrounds remain unknown[9,10]. Type-Ia supernovae emit MeV gamma-rays through nuclear decay line emission[11,12]. However, the event rate estimated by recent observations revealed that the expected signal is below the measured background[13,14]. Kilonovae, transients powered by r-process nuclei ejected by neutron star mergers, also emit MeV gamma-rays through nuclear decay line emission, but the predicted flux is not enough to explain the data[15]. Radio-loud AGN[16] including blazars[17] are possible candidate sources, where nonthermal electrons accelerated in their jets emit MeV gamma-rays. However, recent population studies using *Swift* Burst Alert Telescope (BAT) data suggest that they can contribute to only a part of the measured MeV background[18–20]. Alternatively, Refs. [21,22] suggested that nonthermal electrons accelerated in the coronae of radio-quiet AGN can account for the MeV gamma-ray background. However, the existence of nonthermal electrons is in question, because electrons are rapidly thermalized by Coulomb collisions in typical coronae. On the other hand, proton acceleration by magnetic reconnection and plasma turbulence may occur due to their slower Coulomb relaxation, from which hadronic gamma-rays are unavoidable through proton-induced electromagnetic cascades inside the magnetized coronae[23]. In this case, radio-quiet AGN can explain at least ~ 20% of the observed MeV gamma-ray background, if AGN coronae account for the neutrino data in the 10–100 TeV range.

Here, we propose a scenario that naturally explains the soft gamma-ray background in the MeV range without relying on nonthermal mechanisms. Hot accretion flows, or radiatively inefficient accretion flows (RIAFs)[24,25], are widely expected in low-luminosity AGN (LLAGN), where the feature of an optically thick accretion disk, namely the big-blue bump, is not seen in their optical/UV spectra[26]. Synchrotron radiation from hot thermal electrons explains the observed infrared and radio emission, and the associated soft gamma-rays from Comptonization naturally make a significant contribution to the gamma-ray background up to ~3 MeV. In addition, protons in RIAFs may be accelerated and efficiently emit neutrinos via hadronic interactions[27], which implies that our RIAF model can account for the MeV gamma-ray and neutrino backgrounds simultaneously. A combination of nonthermal protons and thermal electrons in RIAFs is naturally expected. Protons are collisionless in the sense that the relaxation timescale is longer than the dissipation timescale, whereas electrons are rapidly thermalized within the dissipation timescale via synchrotron-self absorption and Coulomb collisions[28].

## Results

**Guaranteed soft gamma rays from RIAFs.** First, we describe the properties of the thermal plasma in RIAFs (see subsection Emission from thermal electrons in RIAFs in Methods and Ref. [29] for technical details). Thermal electrons in RIAFs emit broadband photons through synchrotron radiation, bremsstrahlung, and inverse Compton scattering. For the parameter set shown in Table 1, values of the magnetic field, $B$, the Thomson optical depth, $\tau_T = n_p \sigma_T R$ ($n_p$ is number density, $\sigma_T$ is Thomson cross section, and $R$ is the size of accreting plasma), and the normalized electron temperature, $\Theta_e = k_B T_e/(m_e c^2)$ ($m_e$ is electron mass, $k_B$ is

Boltzmann constant, and $c$ is speed of light), are tabulated in Table 2. RIAFs are optically thick to synchrotron self-absorption at the synchrotron characteristic energy, $\varepsilon_{\rm syn,ch} \approx 3 h_p e B \Theta_e^2/(4\pi m_e c)$, where $h_p$ is Planck constant and $e$ is the elementary charge. The absorption energy above which RIAFs become optically thin is determined by equating the synchrotron emissivity to the blackbody radiation: $\varepsilon_{\rm syn,ab} \approx 3 x_M h_p e B \Theta_e^2/(4\pi m_e c) \simeq 7.0 \times 10^{-3} (B/0.18\,{\rm kG})(\Theta_e/1.5)^2 (x_M/10^3)$ eV, where $x_M = \varepsilon_{\rm syn,ab}/\varepsilon_{\rm syn,ch} \sim 10^3$ represents the correction from $\varepsilon_{\rm syn,ch}$ to $\varepsilon_{\rm syn,ab}$. See Ref. [30] for the values and the method to estimate $x_M$. The synchrotron luminosity from an optically thick source is given by

$$L_{\rm syn} \approx \frac{4 \varepsilon_{\rm syn,ab}^3 k_B T_e}{h_p^3 c^2} \pi^2 R^2 \simeq 2.3 \times 10^{40} \text{ erg s}^{-1}$$
$$\left(\frac{M}{10^8 M_\odot}\right)^2 \left(\frac{\mathcal{R}}{10}\right)^2 \left(\frac{B}{0.18\,{\rm kG}}\right)^3 \left(\frac{\Theta_e}{1.5}\right)^7 \left(\frac{x_M}{10^3}\right)^3, \quad (1)$$

where $M$ is the mass of the supermassive black hole (SMBH), $\mathcal{R} = R/R_S$, and $R_S = 2GM/c^2$ is the Schwarzschild radius. The thermal electrons up-scatter the synchrotron photons, and the resulting spectrum can be approximated by a power-law form, $\varepsilon_\gamma L_{\varepsilon_\gamma} \propto \varepsilon_\gamma^{1-\alpha_{\rm IC}}$, where $\alpha_{\rm IC} = -\ln \tau_T/\ln A_{\rm IC}$ and $A_{\rm IC} = 4\Theta_e + 16\Theta_e^2$[31]. If $\alpha_{\rm IC} < 1$, the Comptonization dominates over other cooling processes, which is satisfied for $\dot{m} \gtrsim 2 \times 10^{-3}$ as seen in Table 2. In this case, the luminosity of the Comptonized photons is

$$L_{\rm IC} \approx L_{\rm syn} \left(\frac{3 k_B T_e}{\varepsilon_{\rm syn,ab}}\right)^{1-\alpha_{\rm IC}} \propto \Theta_e^{6+\alpha_{\rm IC}}. \quad (2)$$

Owing to the strong $\Theta_e$ dependence, our model predicts RIAF electron temperatures which lie in a narrow range $\Theta_e \sim 1 - 3$, unavoidably leading to a peak in the MeV range for $\dot{m} \gtrsim 2 \times 10^{-3}$. The normalization is determined by the balance with the

**Table 1 Model parameters in our reference model.**

| $\alpha$ | $\beta$ | $\mathcal{R}$ | $\eta_{\rm rad,sd}$ | $M[M_\odot]$ | $\kappa_{\rm bol/X}$ | $\kappa_{\rm X/H\alpha}$ | $\eta_{\rm CR}$ | $q$ | $\eta_{\rm tur}$ |
|---|---|---|---|---|---|---|---|---|---|
| 0.1 | 7.0 | 10 | 0.1 | $1 \times 10^8$ | 15 | 6.0 | $4 \times 10^{-4}$ | 1.66 | 15 |

$\alpha$ is the viscous parameter, $\beta$ is the plasma beta, $\mathcal{R}$ is the normalized radius of the RIAF, $\eta_{\rm rad,sd}$ is the radiation efficiency for a standard disk, $M$ is the mass of the SMBH, $\kappa_{\rm bol/X}$ is the correction factor from the X-ray to bolometric luminosities, $\kappa_{\rm X/H\alpha}$ is the correction factor from H$\alpha$ to X-ray luminosities, $\eta_{\rm CR}$ is the CR production efficiency, $q$ is the power-law index of the turbulence power spectrum, and $\eta_{\rm tur}$ is the turbulence parameter.

**Table 2 Resulting quantities for various $\dot{m}$ in our models.**

| log $\dot{m}$ | $B$ | log $\tau_T$ | $\Theta_e$ | $\alpha_{\rm IC}$ | log $\varepsilon_{\gamma\gamma}$ | log $L_{\rm H\alpha}$ | $P_{\rm CR}/P_{\rm th}$ |
|---|---|---|---|---|---|---|---|
| | [kG] | | | | [MeV] | [erg s$^{-1}$] | [%] |
| −3.33 | 0.038 | −2.38 | 3.18 | 1.06 | 5.57 | 38.00 | 2.05 |
| −2.88 | 0.064 | −1.93 | 2.62 | 0.92 | 5.19 | 38.90 | 1.89 |
| −2.43 | 0.11 | −1.48 | 2.01 | 0.79 | 4.06 | 39.81 | 1.60 |
| −1.98 | 0.18 | −1.02 | 1.46 | 0.63 | 3.28 | 40.72 | 1.18 |
| −1.52 | 0.30 | −0.57 | 1.04 | 0.42 | 2.06 | 41.62 | 0.78 |

$\dot{m}$ is the normalized mass accretion rate, $B$ is the magnetic field, $\tau_T$ is the Thomson optical depth, $\Theta_e$ is the normalized electron temperature, $\alpha_{\rm IC}$ is the spectral index of Comptonized photons, $\varepsilon_{\gamma\gamma}$ is the cutoff energy of photons by $\gamma\gamma$ interactions, $L_{\rm H\alpha}$ is the H$\alpha$ luminosity, $P_{\rm CR}/P_{\rm th}$ is the ratio of CR pressure to thermal one.

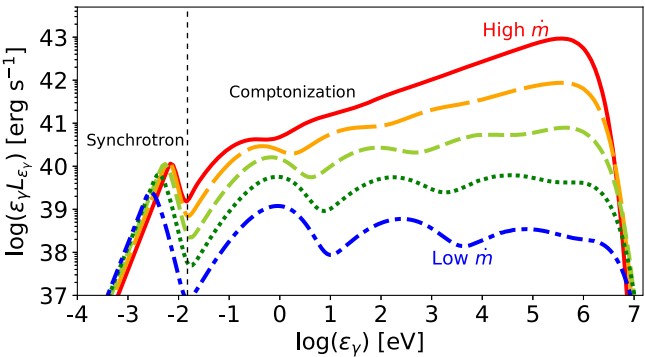

**Fig. 1 Broadband photon spectra from thermal electrons in RIAFs.** We use the parameter set for model A (reference model) with $M = 10^8 M_\odot$ and various $\dot{m}$. The solid, long-dashed, short-dashed, dotted, and dotted-dashed lines are for $\dot{m} = 0.03$, $1.1 \times 10^{-2}$, $3.7 \times 10^{-3}$, $1.3 \times 10^{-3}$, $4.6 \times 10^{-4}$, respectively. The photons of energies below the vertical dotted line are mainly emitted by the synchrotron process, while the photons above the energy are produced by the Comptonization process.

Coulomb heating:

$$L_{IC} \approx (1 - \alpha_{IC})L_{bol} \simeq 1.4 \times 10^{42} \text{erg s}^{-1}$$

$$\left(\frac{M}{10^8 \, M_\odot}\right)\left(\frac{\dot{m}}{0.01}\right)^2\left(\frac{\eta_{rad,sd}}{0.1}\right)\left(\frac{\alpha}{0.1}\right)^2\left(\frac{1-\alpha_{IC}}{0.33}\right), \quad (3)$$

where $\dot{m}$ is the normalized accretion rate, $\eta_{rad,sd}$ is radiation efficiency for a standard disk, and $\alpha$ is the viscous parameter (See subsection Emission from thermal electrons in RIAFs in Methods for the Coulomb heating rate and definition of some parameters).

The broadband photon spectra from thermal electrons are shown in Fig. 1 for various values of $\dot{m}$. The synchrotron emission produces a peak at 0.001–0.01 eV depending on $\dot{m}$, and the Comptonization of the synchrotron photons creates higher-energy photons up to 1–10 MeV. Cases with higher $\dot{m}$ have harder spectra because of their higher Thomson optical depths (see also Refs. [32,33]), making their spectral peaks in the MeV range. These features are quantitatively consistent with the analytic estimates in Equations (1) and (3).

Our model is consistent with observations of nearby LLAGN. Ref. [34] reported a softening feature in the hard X-ray band in NGC 3998, from which they claimed that the electron temperature is $\simeq 30$–40 keV. Our RIAF model can reproduce the softening feature in the NuSTAR band, as well as the *Swift* BAT data shown in Fig. 2, despite a higher electron temperature. Ref. [34] also provided the X-ray spectrum for NGC 4579, which has a higher $\dot{m}$ and does not show any softening feature. Our model also produces a hard power-law spectrum consistent with the NuSTAR data (see Fig. 2). In our RIAF model, the resulting spectra for NGC 3998 and NGC 4579 are relatively hard, and well below the longer wavelength data (radio, infrared/optical/ultraviolet, and soft X-rays). These should be attributed to other emission components, such as compact jets or outer accretion disks[35]. Indeed, radio jets are observed in both objects[36,37].

The Event Horizon Telescope Collaboration (EHT) reported a horizon-scale image of the SMBH in M87[38]. Its brightness temperature is $\sim 5 \times 10^9$ K (equivalent to $\Theta_e \sim 0.8$), while the real temperature should be $\Theta_e \simeq 3$–10, because the image is beam-smeared and the RIAF is likely optically thin at the observed frequency. Our model predicts $\Theta_e \simeq 3.5$ with the parameters appropriate for M87 ($M = 6.3 \times 10^9 M_\odot$, $\dot{m} = 6.1 \times 10^{-4}$), which matches the expected temperature. The electron temperature in RIAFs also affects the interpretation of the photon ring observed by EHT[38,39]. The emission region of the photon ring is

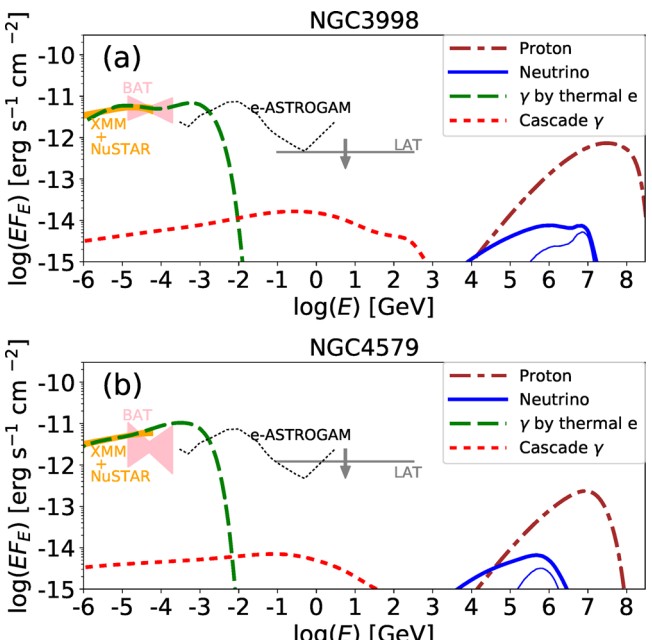

**Fig. 2 Spectra for various particles from nearby LLAGN.** The data by *XMM-Newton* & NuSTAR (orange regions; with a systematic error of 10%), *Swift* BAT (pink regions with 90% confidence levels), and *Fermi* LAT (downward arrows; upper limits with 95% confidence levels) are obtained from Ref. [34], Ref. [112], and Ref. [40], respectively. **a** Spectra for photons from thermal electrons (dashed lines), nonthermal protons (dotted-dashed), total neutrinos (thick-solid), $p\gamma$ neutrinos (thin-solid), and photons by electromagnetic cascades (thick dotted) for NGC 3998. We use $M = 8.1 \times 10^8 M_\odot$[113], $\dot{m} = 2.1 \times 10^{-3}$, and $D_L = 14.1$ Mpc. The thin-dotted line is the sensitivity curve of e-ASTROGAM with 1-yr integration[41]. **b** Same as (**a**), but for NGC 4579. We use $M = 7.2 \times 10^7 M_\odot$[101], $\dot{m} = 8.0 \times 10^{-3}$, and $D_L = 16.4$ Mpc. The NuSTAR data is not smoothly connected to the BAT data, and given the huge statistical error bars in the BAT data, we ignore the BAT data.

determined by the electron temperature and magnetization, which should be clarified through the future multi-wavelength modeling of nearby LLAGN.

**Nonthermal particles in RIAFs.** Protons in RIAFs are accelerated by magnetohydrodynamic (MHD) turbulence and/or magnetic reconnection generated by the magnetorotational instability (MRI). Here, we focus on the stochastic proton acceleration mechanism, in which nonthermal particles randomly gain or lose their energy via interactions with turbulent MHD waves. The accelerated protons, or cosmic-rays (CRs), produce neutrinos and gamma-rays via hadronuclear and photohadronic interactions with thermal protons and photons inside the RIAFs. Neutrinos freely escape from the system, whereas the gamma-rays create electron/positron pairs via $\gamma\gamma \to e^+ e^-$, which initiates proton-induced electromagnetic cascades. Figure 2 shows the resulting proton, neutrino, and proton-induced cascade gamma-ray spectra for NGC 3998 and NGC4579. The proton spectrum is hard because of the stochastic acceleration and has a cutoff around 10–100 PeV due to photohadronic interactions. The neutrinos are mainly produced by $pp$ interactions for $\varepsilon_\nu \lesssim 10^5$–$10^6$ GeV, where $\varepsilon_\nu$ is the neutrino energy, while $p\gamma$ interactions are more efficient around the cutoff energy (See subsection Nonthermal particles in RIAFs in Methods for details). The resulting cascade gamma-ray spectrum is flat for $\varepsilon_\gamma < \varepsilon_{\gamma\gamma}$, where $\varepsilon_\gamma$ is the gamma-ray energy and $\varepsilon_{\gamma\gamma}$ is the energy above which gamma-rays are efficiently attenuated. The gamma-ray flux decreases rapidly

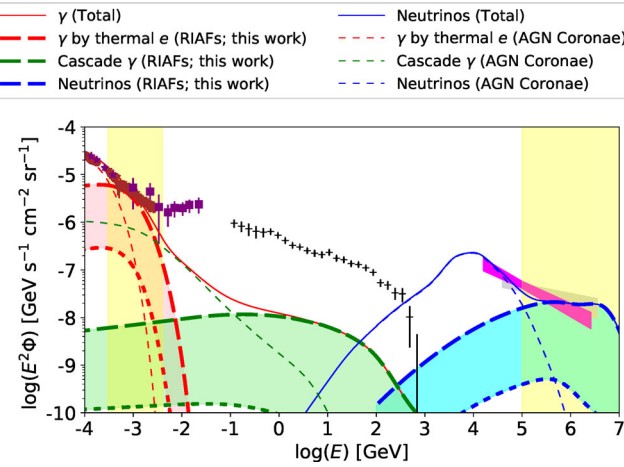

**Fig. 3 Gamma-ray and all-flavor neutrino background intensities.** Data points are provided by *Swift* BAT[115] (brown-circle; 1-σ errors for systematic and statistical errors), SMM[116] (brown-triangle; definition of errors are not available), Nagoya-baloon[117] (purple-star; definition of errors are not available), COMPTEL[2] (purple-square; 2-σ error bars for the linear sum of the systematic and statistical errors), *Fermi* LAT[3] (black-plus; 1-σ errors with systematic and statistical uncertainties), and IceCube[4,118] (lightgrey and magenta regions; 68% confidence levels). The red, blue, and green lines indicate background intensities for photons by thermal electrons, neutrinos by CR protons, and gamma-rays by proton-induced electromagnetic cascades, respectively. The thick, thin-dotted, and thin-solid lines are contributions from RIAFs in LLAGN (this work), coronae in luminous AGN[23], and sum of these, respectively. Thick-dashed and thick-dotted lines are drawn with the luminosity functions of Ref. [45] and Ref. [53], respectively, and the regions between the two lines are filled with the corresponding colors. The yellow vertical bands represent the energy bands where LLAGN provide the dominant contribution. The *Fermi* data should be reproduced by another source class, such as radio-loud AGN[7,52,114].

above the energy. This feature is commonly seen in photon spectra from well-developed electromagnetic cascades[23]. The cascade gamma-ray spectra are well below the upper limit from the *Fermi* Large Area Telescope (LAT)[40] and the design sensitivity of future MeV satellites[41–44].

**Cosmic gamma-ray and neutrino backgrounds**. We calculate the gamma-ray and neutrino background intensities from the RIAFs in LLAGN (see subsection Cumulative background intensities in Methods for details). Fig. 3 shows the resulting gamma-ray and neutrino intensities from LLAGN. In our RIAF model, the Comptonized emission from the thermal electrons naturally accounts for the soft gamma-ray background in the 0.3–3 MeV range, below which canonical AGN coronae explain the X-ray background. Furthermore, nonthermal protons in RIAFs can simultaneously reproduce the IceCube data above ~0.3 PeV, below which hot coronae in luminous AGN can account for the neutrino data in the 10–100 TeV range. Hence, our synthesized AGN core scenario provides an attractive unified explanation for a wide energy range of the cosmic keV–MeV photon and TeV–PeV neutrino backgrounds, as demonstrated in Fig. 3. Notably, AGN accretion flows can do this using a reasonable set of plasma parameters of the nonthermal proton component: $\beta \sim 1$–10, $\eta_{tur} \sim 10$–20, and $P_{CR}/P_{th} \sim 0.01$[23] (see Table 2). This value of $P_{CR}/P_{th}$ is reasonable in the sense that the CR energy density is lower than the magnetic field energy density. Also, the parameters related to nonthermal proton production, such as $\eta_{tur}$ and $P_{CR}/P_{th}$, are expected to be similar in both AGN coronae and RIAFs, because they share the same CR acceleration and

turbulence generation mechanisms. In this sense, the parameters of the nonthermal particles in our RIAF model are effectively calibrated by the 10–100 TeV neutrino data and the AGN corona model.

As shown in panel (a) of Fig. 4, it is the relatively more luminous LLAGN that mainly contribute to the MeV intensity. In particular, LLAGN with $L_{H\alpha} \sim L_{crit}$ provide the dominant contribution, where $L_{crit}$ is the Hα luminosity for $\dot{m} = \dot{m}_{crit}$, where $\dot{m}_{crit}$ is the critical mass accretion rate above which RIAFs no longer exist (See subsection Emission from thermal electrons in RIAFs in Methods for the value of $\dot{m}_{crit}$). Thus, we can analytically estimate the gamma-ray background intensity to be[23]

$$E_\gamma^2 \Phi_\gamma \sim \frac{c}{4\pi H_0} \xi_z \rho_* \left(\frac{L_*}{L_{crit}}\right)^{s_1-1} \varepsilon_\gamma L_{\varepsilon_\gamma}$$

$$\sim 3 \times 10^{-6}\, \mathrm{GeV\, s^{-1}\, cm^{-2}\, sr^{-1}} \left(\frac{\xi_z}{0.6}\right) \left(\frac{\varepsilon_\gamma L_{\varepsilon_\gamma}}{45 L_{crit}}\right) \left(\frac{\dot{m}_{crit}}{0.03}\right)^{-0.05}, \quad (4)$$

where $H_0 \sim 70\, \mathrm{km\, s^{-1}\, Mpc^{-3}}$ is the Hubble constant, $L_*$, $\rho_*$, and $s_1$ are the break luminosity, break density, and power-law index for the luminosity function[45] (see subsection Cumulative background intensities in Methods), respectively, and $\xi_z$ is the redshift evolution factor of the luminosity density. This is consistent with our numerical results and the observed MeV background from COMPTEL. We stress that the resulting MeV gamma-ray intensity does not depend strongly on the parameters related to the RIAF, such as $\alpha$, $B$, $M$, $\dot{m}_{crit}$, nor on the electron heating prescription, as long as we can use the luminosity function and bolometric correlation. This is because LLAGN of $\dot{m} \sim \dot{m}_{crit}$ provide the dominant contribution, and the energy budget of such LLAGN does not strongly depend on the value of $\dot{m}_{crit}$.

On the other hand, the relatively faint LLAGN mainly contribute to the neutrino background (see panel (b) of Fig. 4). The high-energy gamma-rays accompanying the neutrinos are considerably attenuated inside RIAFs (see Table 2 for values of the break energy due to $\gamma\gamma \to e^+e^-$, $\varepsilon_{\gamma\gamma}$), making the GeV gamma-ray intensity well below the *Fermi* data. Thus, LLAGN can be regarded as gamma-ray hidden neutrino sources[46].

Our RIAF model provides a ~5–10% contribution to the cosmic soft X-ray (0.5–8 keV) background, which is dominated by canonical AGN and star-forming galaxies[6,47]. In observations of distant LLAGN, fainter ones are likely to be classified as star-forming galaxies, while relatively luminous ones will be indistinguishable from the faint end of canonical AGN. Thus, it is difficult to determine the contribution by LLAGN accurately. Here, we make a rough estimate of the LLAGN contribution. The luminous objects of $L_X > 3.2 \times 10^{42}\mathrm{erg\, s^{-1}}$ provides 63% of the CXB flux. Our LLAGN should have X-ray luminosities below this range, and thus, 37% contribution can be regarded as an upper limit for the LLAGN contribution. Ref. [26] reported that 43% of nearby galaxies have signatures of AGN activity. Then, the LLAGN contribution can be as high as $0.43 \times 0.37 \simeq 0.16$. Our model predicts a lower LLAGN contribution than our rough estimate. Future X-ray and optical spectroscopic surveys with better sensitivities are necessary to unravel the X-ray contribution from LLAGN.

**Tests by future observations**. High-energy multimessenger observations are essential for identifying the origin of the gamma-ray and neutrino backgrounds. We have estimated the detectability of neutrinos from RIAFs (see subsection Neutrino detectability from nearby LLAGN in Methods for details). The expected number of through-going muon track events from nearby LLAGN, $\mathcal{N}_\mu$, is shown in Fig. 5. Current facilities are not sufficient to detect the signal, even using stacking techniques. However, stacking ~10 LLAGN will enable the planned

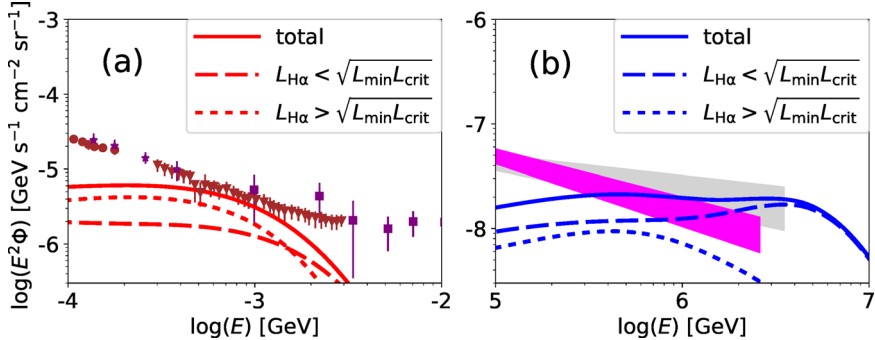

**Fig. 4 Contributions by relatively luminous and faint LLAGN. a** Diffuse soft gamma-ray intensities from relatively luminous (dotted) and faint (dashed) LLAGN. The solid line shows the sum of these. Data points are provided by *Swift* BAT[115] (brown-circle; 1-$\sigma$ errors for systematic and statistical errors), SMM[116] (brown-triangle; definition of errors are not available), Nagoya-baloon[117] (purple-star; definition of errors are not available), COMPTEL[2] (purple-square; 2-$\sigma$ error bars for the linear sum of the systematic and statistical errors). **b** Same as (**a**) but for diffuse neutrino intensities. Data are provided by IceCube[4,118] (lightgrey and magenta regions; 68% confidence levels).

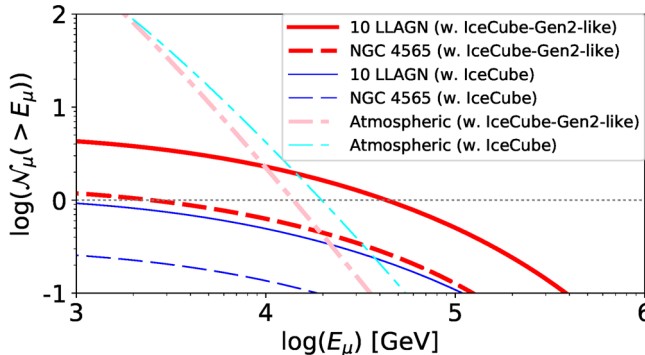

**Fig. 5 The expected number of neutrino events with current and future detectors.** We plot the expected number of through-going muon track events above a given muon energy for a 10-yr operation with IceCube (thin-blue) and IceCube-Gen2-like detector (thick-red). The solid lines are for the case that stacks the ten brightest LLAGN, and the dashed lines are for the brightest LLAGN, NGC 4565. The dotted-dashed lines show the expected number of the atmospheric background for the case that stacks the ten LLAGN. The dotted horizontal line indicates $\mathcal{N}_\mu = 1$ for comparison. Stacking more LLAGN does not help improving the detectability because the background also increases with the number of the stacked LLAGN[29].

IceCube-Gen2 facility to detect a few neutrino events by means of its larger effective area. Its better angular resolution will reduce the atmospheric background, making individual source detections possible.

Future MeV satellites, such as e-ASTROGAM[41], the All-sky Medium Energy Gamma-ray Observatory (AMEGO)[48], and Gamma-Ray and AntiMatter Survey (GRAMS)[43], will be able to measure the electron temperature in RIAFs by detecting a clear cutoff feature (see Fig. 2), which will shed light on the electron heating mechanisms in the collisionless plasma. In addition, anisotropy tests for the MeV gamma-ray background are promising[44]. Our RIAF model predicts a smaller anisotropy than those from Seyfert and blazar models owing to the higher source number density of LLAGN.

**Possible effects of another acceleration mechanism.** If protons are accelerated by another mechanism, such as magnetic reconnection, the resulting proton spectrum is usually described by a power-law form. In the upper panel of Fig. 6, we plot the diffuse neutrino and gamma-ray intensities for power-law injection models with the parameter sets tabulated in Table 3 (see subsection Power-law injection models for CRs in Methods).

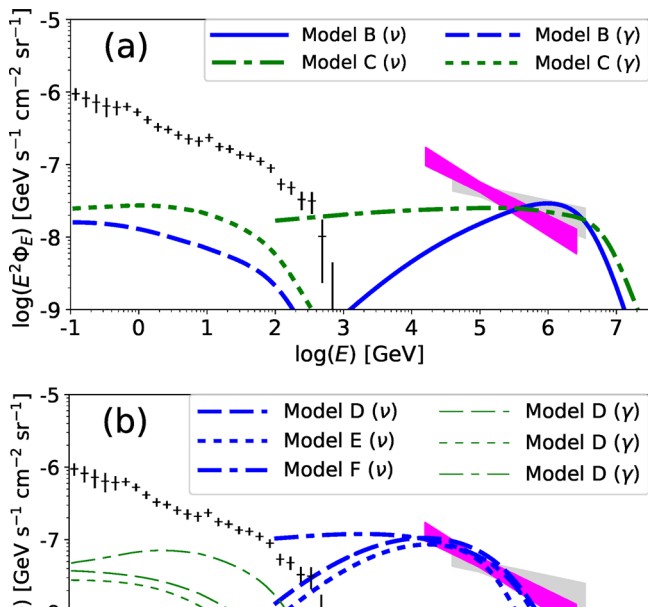

**Fig. 6 All-flavor neutrino and gamma-ray background intensities for power-law injection models and models for 10–100 TeV neutrinos.** Data are provided by *Fermi* LAT[3] (black-plus; 1-$\sigma$ errors with systematic and statistical uncertainties), and IceCube[4,118] (lightgrey and magenta regions; 68% confidence levels). **a** Power-law injection models that can account for the PeV neutrino data. The blue and green lines are for models B ($s_{inj} = 1$) and C ($s_{inj} = 2$) in Table 3, respectively. The solid and dotted-dashed lines indicate the neutrino spectra, and the dashed and dotted lines represent the gamma-ray spectra by proton-induced electromagnetic cascades. **b** Models for 10–100 TeV neutrinos. The thick-blue and thin-green lines are for neutrinos and proton-induced cascade gamma-rays, respectively. The dashed, dotted, dotted-dashed lines are for models D, E, F in Table 3, respectively.

The power-law injection models can also account for the PeV neutrino data with $P_{CR}/P_{th} \sim 0.01$ for model B and $P_{CR}/P_{th} \sim 0.1$ for model C. A higher $P_{CR}/P_{th}$ is demanded for a higher $s_{inj}$, which may lead some feedback to the MHD turbulence (see discussion in the next subsection). The cascade gamma-ray emissions are well below the *Fermi* data owing to the low $\gamma\gamma$ break

**Table 3 Parameters for models for PeV neutrinos (A, B, C) and 10–100 TeV neutrinos (D, E, F).**

| Parameters | $\eta_{CR}[10^{-3}]$ | $\eta_{tur}$ | $s_{inj}$ | $\eta_{acc}[10^4]$ |
|---|---|---|---|---|
| Model A (reference) | 0.40 | 15 | – | – |
| Model B | 0.40 | – | 1.0 | 2.0 |
| Model C | 2.0 | – | 2.0 | 0.50 |
| Model D | 3.0 | 50 | – | – |
| Model E | 2.0 | – | 1.0 | 70 |
| Model F | 10 | – | 2.0 | 15 |

$\eta_{CR}$ is the CR production efficiency, $\eta_{tur}$ is the turbulent strength, $s_{inj}$ is the spectral index of CR injection term, and $\eta_{acc}$ is the acceleration efficiency parameter.

energy in the RIAFs. Note that the resulting diffuse MeV gamma-ray intensities for models B and C are the same as that for model A, because the thermal electrons in the RIAFs are identical.

**Medium-energy neutrinos in the 10–100 TeV range**. For the sake of completeness, we also investigate the possibility of whether our model can reproduce the 10–100 TeV neutrino background. The observed intensity of the 10–100 TeV neutrinos is higher than that of the 100 GeV gamma-rays. Hence, the emission region of origin of the 10–100 TeV neutrinos is expected to be opaque to GeV–TeV gamma-rays[46].

The lower panel of Fig. 6 depicts the diffuse gamma-ray and neutrino intensities for models D (stochastic acceleration), E (power-law injection with $s_{inj} = 1$), and F (power-law injection with $s_{inj} = 2$), whose parameters are tabulated in Table 3. Since the thermal component is identical to that for model A, we only show the nonthermal components.

The neutrino intensity is normalized so that it matches the observed 10–100 TeV neutrinos, which requires higher values of $\eta_{CR}$ and $\eta_{tur}$ or $\eta_{acc}$. The gamma-ray intensity from proton-induced cascades is also higher than that of PeV neutrinos, but still considerably lower than the *Fermi* data. Therefore, our LLAGN model could in principle be the source of the mysterious 10–100 TeV neutrinos, although the contribution from Seyferts and quasars is typically more important in view of the energetics.

The resulting pressure ratio of the CRs to the thermal component is about 10% for models D and E and 50% for model F. These are much higher than those for models for PeV neutrinos. Nevertheless, even with such a high value of $P_{CR}/P_{th}$, the global dynamical structure of RIAFs are very similar as long as CRs are confined inside the flow as shown in Ref. [49]. However, such a high value of $P_{CR}/P_{th}$ is at odds with the picture of turbulent acceleration, and the feedback from CRs to MHD turbulence may be significant. In Ref. [29], we estimated the point-source detectability of the neutrinos from nearby LLAGN using the method in Ref. [50] with the parameter sets similar to those for models D, E, and F. The diffuse neutrino spectra for the models in Ref. [29] are almost identical to those for models D, E, and F. Ref. [29] showed that the TeV–PeV neutrinos and MeV gamma-rays are detectable with the planned neutrino and gamma-ray experiments, IceCube-Gen2 and AMEGO/e-ASTROGAM/GRAMS, respectively.

## Discussion

The present work differs from earlier works in a number of respects, as described below. Ref. [23] calculated neutrino and gamma-ray emission from hot coronae surrounding the accretion disks in luminous AGN, i.e., Seyfert galaxies and quasars. The stochastic proton acceleration was considered, and proton-induced electromagnetic cascades are fully taken into account.

It was shown that the high-energy emission from coronae can account for the X-ray and 10–100 TeV neutrino backgrounds that require gamma-ray hidden sources. Note that the structure of the systems considered in the present work and Ref. [23] are different. The present work considers the RIAFs in LLAGN, where CRs can be accelerated in the bulk of the accretion flows. A broadband of target photons are provided by the thermal electrons inside the RIAFs. Ref. [23] used the disk-corona paradigm in luminous AGN, where protons are accelerated at the hot corona above the geometrically thin, optically thick standard disk. The disks provide thermal UV photons, while the coronae supplies X-rays by upscattering the UV photons. Strictly speaking, the prescription of the nonthermal particle injection for our RIAF model is also different from that in Ref. [23]. In the RIAF model, we use a constant $\eta_{CR}$ parameter, i.e., the fraction of CR luminosity to the accretion luminosity is independent of $\dot{m}$. On the other hand, in the AGN corona model, we considered that $\dot{\mathcal{F}}_{p,inj}$ is proportional to $L_X$, where $\eta_{CR}$ depends on $L_X$ and $\dot{m}$. The injection process of nonthermal CR acceleration remains an open problem, so we examined effects of both prescriptions. We find that our RIAF and the AGN corona models can reproduce 0.1–1 PeV and 10–100 TeV neutrino datas, respectively, with either of the prescriptions, with a similar value of $P_{CR}/P_{th} \sim 0.01$. Our conclusions are nearly independent of these prescriptions about the injection. Ref. [27] calculated the neutrino background from RIAFs, considering stochastic particle acceleration by MHD turbulence. It considered neither gamma-ray emission nor neutrino point-source detectability, but studied the neutrino background for either 10–100 TeV or 0.1–1 PeV components. Ref. [29] focused on neutrino and gamma-ray emissions from nearby LLAGN with RIAFs, and discussed the neutrino and gamma-ray detectabilities in future experiments. In addition to the stochastic acceleration model, the power-law injection models were discussed. Ref. [51] considered gamma-ray and neutrino emission from magnetically arrested disks (MADs) in radio galaxies, motivated by *Fermi* detection of nearby radio galaxies. They assume an efficient acceleration close to the theoretical limit, which could be possible by magnetic reconnections. The maximum energy of the accelerated particles is much higher than in the other papers on luminous AGN and LLAGN, leading to efficient GeV gamma-ray production by the proton synchrotron process.

The H$\alpha$ luminosity function has uncertainties. Ref. [52] proposed a lower value of H$\alpha$ luminosity function. Then, an analytic estimate using Eq. (4) gives $E_\gamma^2 \Phi_\gamma \sim 7 \times 10^{-7}$ GeV s$^{-1}$ cm$^{-2}$ sr$^{-1}$, which is an order of magnitude lower than the MeV gamma-ray data as shown in Fig. 3. Even with the lower value of the luminosity function, the PeV neutrino data may be reproduced if we use $\eta_{CR} \sim 0.02$, which leads to a high CR energy density, $P_{CR}/P_{th} \sim 0.5$. However, such a situation is at odds with the assumption that the turbulence energy is the source of the CRs. The uncertainty in the luminosity function should be palliated by future surveys using a line-sensitive optical instrument or sensitive X-ray satellites, such as Subaru Prime Focus Spectrograph[53] or extended Roentgen Survey with an Imaging Telescope Array (eROSITA)[54] and Focusing On Relativistic universe and Cosmic Evolution (FORCE)[55].

The $e^+e^-$ pair production processes are inefficient in RIAFs. The most efficient process is the two-photon ($\gamma\gamma$) interaction. The optical depth for MeV photons to the $e^+e^-$ pair production is estimated to be $\tau_{\gamma\gamma} \approx n_t \sigma_{\gamma\gamma} R \simeq 1.4 \times 10^{-3} L_{IC,42} R_{14.5}^{-1}$, where $\sigma_{\gamma\gamma} \sim 0.2\sigma_T$ is the cross section for two-photon interactions, $n_t \sim L_{IC}/(4\pi R^2 m_e c^3) \sim 3.2 \times 10^7 L_{IC,42} R_{14.5}^{-2}$ cm$^{-3}$ is the target photon density at MeV energies, and we use convention of $Q_X = Q/10^X$ in cgs unit. Equating the $e^+e^-$ pair production rate and advective escape rate, the number density of the electron-positron

pairs is estimated to be[51] $n_{\pm} \approx n_t \tau_{\gamma\gamma}(c/V_R) \sim 4 \times 10^6$ $L_{IC,42}^2 R_{14.5}^{-5/2} \alpha_{-1}^{-1}$ cm$^{-3}$, which is a few orders of magnitude smaller than the electron density given by Eq. (6). Electron-electron (ee) and electron-proton (ep) interactions can also produce $e^+e^-$ pairs, whose timescales are roughly approximated to be $t_{ep,\pm} \sim t_{ee,\pm} \sim 1/(n_p \alpha_{em}^2 \sigma_T c)$, where $\alpha_{em}$ is the fine structure constant. Then, the ratio of the $e^+e^-$ pair production timescale to the infall timescale is estimated to be $t_{fall}/t_{ee,\pm} \sim 4 \times 10^{-4} \alpha_{-1}^{-2} \dot{m}_{-2}$, and thus, ee and ep interactions cannot provide sufficient amount of pairs. Photon-proton ($\gamma p$) and photon-electron ($\gamma e$) interactions also produce pairs, but the cross sections for these processes at $\varepsilon_\gamma \sim 3$–4 MeV are similar to those for ee and ep interactions, which leads to pair production rates similar to those by ee and ep interactions. Photons with a higher energy have larger $\gamma p$ and $\gamma e$ cross sections, but the number density of such photons is too small to produce the pairs efficiently. Therefore, the RIAF plasma is unlikely to reach the pair equilibrium, in which the density of the electron-positron pairs is much lower than the proton density. Ref. [56] demonstrated this conclusion by detailed calculations.

While this work demonstrates that LLAGN can significantly contribute to the higher-energy part of IceCube neutrinos, contrary to the guaranteed MeV gamma-ray emission, their nonthermal contribution might in principle be much smaller if the CR acceleration in the RIAF disks is more inefficient than in the coronae of radio-quiet AGN. This may be the case if the CR acceleration occurs predominantly in a low-$\beta$ plasma, because $\beta \sim 3$–30 in a RIAF disk is higher than $\beta \lesssim 1$ in AGN coronae. In this case, we would need other models that can explain the neutrino background in the PeV range[57].

In conclusion, we proposed RIAFs in LLAGN as a promising origin for the soft gamma-ray background. We constructed a one-zone model that can reproduce the observed X-ray features of LLAGN, and demonstrated that LLAGN can also simultaneously account for the high-energy neutrino background. In the RIAFs, electrons are thermalized and emit soft gamma rays through Comptonization. The protons there are naturally accelerated by reconnection or turbulence due to their longer thermalization timescale, and produce high-energy neutrinos efficiently. The accompanying gamma-rays are significantly attenuated by two-photon interactions, resulting in a gamma-ray intensity well below the *Fermi* data. Since hot coronae in luminous AGN can produce 10–100 TeV neutrinos through the same mechanism[23], accretion flows in AGN can account for a wide range of high-energy photon (keV–MeV) and neutrino (TeV–PeV) backgrounds. This scenario does not require any nonthermal electron population to account for the MeV background, which implies that the transition energy from the thermal to nonthermal Universe is higher than previously expected.

## Methods

**Emission from thermal electrons in RIAFs.** Here, we describe the properties of thermal plasma in RIAFs[24,58] in detail. Hereafter, we use the notation of $Q_X = Q/10^X$ in cgs unit unless otherwise noted. We consider an accreting plasma of size $R$ around a supermassive black hole (SMBH) of mass $M$ with an accretion rate $\dot{M}$. We use the normalized radius and mass accretion rate, $\mathcal{R} = R/R_S$ and $\dot{m} = \dot{M}c^2/L_{Edd}$, where $R_S$ is the Schwarzschild radius and $L_{Edd}$ is the Eddington luminosity. Plasma quantities in RIAFs are described by two parameters, the viscosity parameter, $\alpha$, and the pressure ratio of gas to magnetic field, $\beta$. Based on recent numerical simulations (e.g., Refs. [59–64]), the radial velocity, number density, proton thermal temperature, magnetic field, Thomson optical depth, and Alfvén velocity in the RIAF are analytically approximated to be

$$V_R \approx \alpha V_K/2 \simeq 3.4 \times 10^8 \mathcal{R}_1^{-1/2} \alpha_{-1} \text{ cm s}^{-1}, \quad (5)$$

$$n_p \approx \frac{\dot{M}}{4\pi m_p R H V_R} \simeq 4.6 \times 10^8 \mathcal{R}_1^{-3/2} \alpha_{-1}^{-1} M_8^{-1} \dot{m}_{-2} \text{ cm}^{-3}, \quad (6)$$

$$k_B T_p \approx \frac{GMm_p}{4R} \simeq 12 \mathcal{R}_1^{-1} \text{ MeV}, \quad (7)$$

$$B \approx \sqrt{\frac{8\pi n_p k_B T_p}{\beta}} \simeq 1.5 \times 10^2 \mathcal{R}_1^{-5/4} \alpha_{-1}^{-1/2} M_8^{-1/2} \dot{m}_{-2}^{1/2} \beta_1^{-1/2} \text{ G}, \quad (8)$$

$$\tau_T \approx n_p \sigma_T R \simeq 0.090 \mathcal{R}_1^{-1/2} \dot{m}_{-2} \alpha_{-1}^{-1}, \quad (9)$$

$$\beta_A \approx \frac{B}{\sqrt{4\pi n_p m_p c^2}} \simeq 0.050 \mathcal{R}_1^{-1/2} \beta_1^{-1/2}, \quad (10)$$

where $V_K = \sqrt{GM/R}$ is the Keplerian velocity and $H \approx R/2$ is the scale height. Observations of X-ray binaries and AGN demand $\alpha \sim 0.1$–1[65], while the global MHD simulations result in $\alpha \sim 0.01$–0.1 and $\beta \sim 3$–30[61,66]. Hence, we set $\alpha = 0.1$ and $\beta = 7.0$ as their reference values.

Thermal electrons in RIAFs emit broadband photons through synchrotron radiation, bremsstrahlung, and inverse Compton scattering. The electron temperature is determined so that the resulting photon luminosity is equal to the bolometric luminosity estimated by $\dot{m}$. Assuming that thermal electrons are heated by Coulomb collisions with protons, the bolometric luminosity is estimated to be $L_{bol} \approx \eta_{rad,sd} \dot{m}_{crit} L_{Edd} (\dot{m}/\dot{m}_{crit})^2$, where $\eta_{rad,sd} \sim 0.1$ is the radiation efficiency for the standard disk, and $\dot{m}_{crit} \approx 0.03(\alpha/0.1)^2$ is the critical mass accretion rate above which RIAFs no longer exist[30]. We calculate the photon spectra by synchrotron radiation, bremsstrahlung, and inverse Compton scattering by thermal electrons with the steady-state and one-zone approximations. The synchrotron and bremsstrahlung spectra are calculated by the method given in Appendix of Ref. [27], where we use the fitting formulae for the emissivity of these processes and Eddington approximation to take into account the effects of the radiative transfer. For the inverse Compton scattering spectrum, we utilize the corrected delta-function method given in Ref. [67]. In this method, the distribution of the scattered photon energy is approximated to be a delta function, and the mean energy of the scattered photon is calculated using the exact kernel. This method approximately takes into account the electron recoil effect for $\varepsilon_\gamma \gtrsim k_B T_e$. The error of the method is about 50%. This is sufficiently accurate for our purpose, considering significant uncertainty in the MeV gamma-ray data. Given the uncertainty, our calculation results are consistent with those by the Monte Carlo simulations[68]. The spectral decline due to the cutoff in this method is somewhat stronger than that in the exact method, implying that our results on the MeV fluxes are regarded as conservative. In this work, we assume that the Coulomb heating is the dominant electron heating mechanism, and $L_{bol} \propto \dot{m}^2$ is used. This treatment is qualitatively different from the previous work[27], where $L_{bol} \propto \dot{m}$ is assumed. Such a treatment may be more appropriate if the electrons are directly heated by the plasma dissipation process[69–72]. Nevertheless, these details will not change our conclusions that LLAGN are bright in soft MeV gamma-rays. Also, the electron heating prescription do not strongly affect the critical mass accretion rate above which RIAFs no longer exist[28,30,73,74].

**Nonthermal particles in RIAFs.** Here, we describe the details of the stochastic acceleration model, where protons are accelerated by MRI turbulence[64,75]. To obtain the CR spectrum, we solve the transport equation for CR protons, which is a diffusion equation in the momentum space[76]:

$$\frac{\partial \mathcal{F}_p}{\partial t} = \frac{1}{\varepsilon_p^2} \frac{\partial}{\partial \varepsilon_p} \left( \varepsilon_p^2 D_{\varepsilon_p} \frac{\partial \mathcal{F}_p}{\partial \varepsilon_p} + \frac{\varepsilon_p^3}{t_{cool}} \mathcal{F}_p \right) - \frac{\mathcal{F}_p}{t_{esc}} + \dot{\mathcal{F}}_{p,inj}, \quad (11)$$

where $\mathcal{F}_p$ is the momentum distribution function for protons $(dN/d\varepsilon_p = 4\pi p^2 \mathcal{F}_p/c)$, $D_{\varepsilon_p}$ is the diffusion coefficient that mimics the stochastic particle acceleration, $t_{cool}$ is the cooling time for protons, $t_{esc}$ is the escape time, $\dot{\mathcal{F}}_{p,inj} = \dot{\mathcal{F}}_0 \delta(\varepsilon_p - \varepsilon_{p,inj})$ is the injection function, and $\varepsilon_{p,inj}$ is the initial energy of the particles injected to the stochastic acceleration process. We assume that the particles are injected to the stochastic acceleration process by fast acceleration processes such as magnetic reconnections, which are induced by MRI[77–79]. We consider a delta-function injection term with $\varepsilon_{p,inj}$ much higher than the thermal proton energy, which may mimic the injection by the reconnection. The value of $\varepsilon_{p,inj}$ has no influence on the resulting spectrum as long as $\varepsilon_{p,inj}$ is much lower than the cutoff energy. We consider resonant scatterings between MHD waves and CR particles, where CR particles interact with the turbulent eddy of their gyration radius, $r_L = \varepsilon_p/(eB)$. Then, diffusion coefficient in energy space can be written as

$$D_{\varepsilon_p} \approx \frac{c\beta_A^2}{\eta_{tur} H} \left( \frac{r_L}{H} \right)^{q-2} \varepsilon_p^2, \quad (12)$$

where $\eta_{tur} = B^2/(8\pi \int P_k dk)$ is the turbulence parameter ($P_k$ is the turbulence power spectrum), $q$ is the power-law index of $P_k$, and we set the scale height, $H$, to the injection scale of the MHD turbulence. We assume a Kolmogorov turbulence, $q = 5/3$. The acceleration time is given by $t_{acc} \approx \varepsilon_p^2/D_{\varepsilon_p} \approx \eta_{tur} \beta_A^{-2} (H/c)[\varepsilon_p/(eBH)]^{1/3}$. $\eta_{tur}$ is lower for a stronger turbulence, and a low value of $\eta_{tur}$ results in a

higher maximum energy of the CR protons. Since RIAFs are expected to be turbulent due to MRI[80,81], the value of $\eta_{tur}$ should be small. The turbulent strength is also related to the value of $\alpha$, as stronger turbulence leads to a higher $\alpha$. Our fiducial value of $\eta_{tur} \sim 15$ is reasonable in the sense that $\eta_{tur}$ is close to $\alpha^{-1}$. The amount of CRs is determined so that $\int L_{\varepsilon_p} d\varepsilon_p = \eta_{CR} \dot{m} L_{Edd}$ is satisfied, where $L_{\varepsilon_p} = t_{loss}^{-1} \varepsilon_p dN_p/d\varepsilon_p$ is the differential proton luminosity[23], $\eta_{CR}$ is the CR production efficiency, and $t_{loss}^{-1} = t_{cool}^{-1} + t_{esc}^{-1}$ is the total energy loss rate including cooling and escape processes.

We solve the transport equation until the steady-state is reached using the Chang-Cooper method[82,83]. As the proton cooling mechanism, we consider the proton synchrotron, Bethe-Heitler ($p + \gamma \rightarrow p + e^+ + e^-$), photomeson ($p + \gamma \rightarrow p + \pi$), and $pp$ inelastic collision ($p + p \rightarrow p + p + \pi$) processes, The timescale of the $pp$ inelastic collisions is given as $t_{pp}^{-1} \approx n_p \sigma_{pp} \kappa_{pp} c$, where $\sigma_{pp}$ and $\kappa_{pp}$ are the cross section and inelasticity of $pp$ interactions[84]. The photomeson production and Bethe-Heitler cooling timescales are estimated to be

$$t_i^{-1} = \frac{c}{2\gamma_p^2} \int_{\bar{\varepsilon}_{th}}^{\infty} d\bar{\varepsilon}_\gamma \sigma_i \kappa_i \bar{\varepsilon}_\gamma \int_{\bar{\varepsilon}_\gamma/(2\gamma_p)}^{\infty} \frac{d\varepsilon_\gamma}{\varepsilon_\gamma^2} n_{\varepsilon_\gamma}, \quad (13)$$

where $\gamma_p = \varepsilon_p/(m_p c^2)$, $\bar{\varepsilon}_\gamma$ is the photon energy in the proton rest frame, and $\sigma_i$ and $\kappa_i$ are the cross section and inelasticity for the process ($i = p\gamma$ for photomeson production[85] and $i = $ BH for Bethe-Heitler process[86,87]. The cooling time by the proton synchrotron is $t_{p,syn} \approx 6\pi m_p^4 c^3/(\sigma_T B^2 \varepsilon_p)$. The total cooling rate is then given by $t_{cool}^{-1} = t_{pp}^{-1} + t_{p\gamma}^{-1} + t_{BH}^{-1} + t_{p,syn}^{-1}$. Regarding the escape process, we only consider the advective escape, i.e., infall to the SMBH. We write the escape timescale as $t_{esc} = t_{adv} \approx R/V_R$. We consider that the CR component has the same bulk velocity as that for the thermal component owing to efficient interactions through the turbulent magnetic field. The diffusive escape may be inefficient because the high-energy protons tend to move in the azimuthal direction, which is the direction of the background magnetic field in differentially rotating accretion flows[64,75]. See also Refs. [23,29] for technical details of the calculation methods for the cooling and escape timescales.

In Fig. 7, we plot the acceleration and loss rates of CRs for model A (reference model) for NGC 4579, whose parameters are shown in Table 1 and caption. The dominant loss process is the advective escape for $\varepsilon_p \lesssim 1 \times 10^8$ GeV. Although the acceleration time is longer than the advective escape time for $\varepsilon_p \gtrsim 2 \times 10^5$ GeV, the CR proton spectrum continues to a higher-energy because the weak $\varepsilon_p$ dependence of the advective escape leads to a very gradual cutoff in the proton spectrum[27,88]. At $\varepsilon_p \sim 1 \times 10^7$ GeV, the photomeson production becomes more efficient than the acceleration, which makes a sharp cutoff owing to a strong $\varepsilon_p$ dependence (see Fig. 2).

The CRs produce pions via both $pp$ and $p\gamma$ interactions, and pions decay to gamma-rays, electron/positron pairs, and neutrinos ($\pi^0 \rightarrow 2\gamma$; $\pi^\pm \rightarrow e^\pm + 3\nu$). These neutrinos are believed to explain IceCube neutrinos[89-91]. We calculate neutrino spectra by $pp$ collisions using the formalism given by Ref. [92]. For the neutrinos by $p\gamma$ interactions, we use a semi-analytic prescription given in Refs. [93,94] (see, e.g., Ref. [85] for numerical results). Since the effect of meson cooling is negligible in the RIAFs, neutrino flavor ratio is $\nu_e:\nu_\mu:\nu_\tau = 1:2:0$ at the source, which becomes $\sim 1:1:1$ on Earth after the flavor mixing. As shown in Fig. 2, neutrinos are mainly produced by the $pp$ collisions for $\varepsilon_\nu \lesssim 4 \times 10^5$ GeV, and the photomeson

production is effective above the energy. The Bethe-Heitler and proton synchrotron processes are subdominant in the range we investigated. For higher $\dot{m}$ cases, the photomeson production is more efficient, and hence, the peak energy of the proton spectrum is lower. In our model, the neutrino background is dominated by relatively faint LLAGN, and their target photon spectra are not hard. Then, the multi-pion production channel is subdominant, and our approximation provides reasonably accurate results.

The hadronic interactions also produce gamma-rays and electron/positron pairs. The gamma-rays are absorbed by two-photon annihilation, and create electron/positron pairs. These pairs also emit gamma-rays, and electromagnetic cascades are initiated. We calculate the cascade emission by solving the kinetic equations of electron/positron pairs and photons[95,96]:

$$\frac{\partial n_{\varepsilon_e}}{\partial t} + \frac{\partial}{\partial \varepsilon_e}\left[ \left( P_{IC} + P_{syn} + P_{ff} + P_{Cou} \right) n_{\varepsilon_e} \right] = \dot{n}_{\varepsilon_e}^{(\gamma\gamma)} - \frac{n_{\varepsilon_e}}{t_{esc}} + \dot{n}_{\varepsilon_e}^{inj}, \quad (14)$$

$$\frac{\partial n_{\varepsilon_\gamma}}{\partial t} = -\frac{n_{\varepsilon_\gamma}}{t_{\gamma\gamma}} - \frac{n_{\varepsilon_\gamma}}{t_{\gamma,esc}} + \dot{n}_{\varepsilon_\gamma}^{(IC)} + \dot{n}_{\varepsilon_\gamma}^{(ff)} + \dot{n}_{\varepsilon_\gamma}^{(syn)} + \dot{n}_{\varepsilon_\gamma}^{inj}, \quad (15)$$

where $n_{\varepsilon_i}$ is the differential number density ($i = e$ or $\gamma$), $\dot{n}_{\varepsilon_i}^{(xx)}$ is the particle source term from the process $xx$ ($xx = $ IC (inverse Compton scattering), $\gamma\gamma$ ($e^+e^-$ pair production by $\gamma\gamma$ interactions), syn (synchrotron), or ff (bremsstrahlung), $\dot{n}_{\varepsilon_i}^{inj}$ is the injection term from the hadronic interaction, and $P_{yy}$ is the energy loss rate for the electrons from the process $yy$ ($yy = $ IC (inverse Compton scattering), syn (synchrotron), ff (bremsstrahlung), or Cou (Coulomb collision). See also Refs. [95,96] for technical details. We approximately treat the pair injection processes by the Bethe-Heitler process and photomeson production as in Refs. [23,29].

In our RIAF models, secondary pairs do not contribute to the the high-energy gamma-ray background, even for the case that the secondary pairs are re-energized by MHD turbulence. The secondary pairs suffer from a strong cooling by the synchrotron emission, whose timescale is estimated to be $t_{e,syn} = 6\pi m_e^2 c^3/(\sigma_T B^2 \varepsilon_e)$. When their energy becomes sufficiently low, they may be re-energized by MHD turbulence, as suggested by Ref. [23]. The energization timescale is the same with the proton acceleration timescale: $t_{e,acc} \approx \eta_{tur} \beta_A^{-2} [H/c][\varepsilon_e/(eBH)]^{2-q}$. Equating these two timescales, we obtain the critical energy at which the secondary pairs piles up:

$$\varepsilon_{e,crit} \approx \left( \frac{6\pi m_e^2 c^4 \beta_A^2}{\sigma_T H \eta_{tur} B^2} \right)^{\frac{1}{3-q}} (eBH)^{\frac{2-q}{3-q}} \simeq 4.2 \, \mathcal{R}_1^{5/16} \alpha_{-1}^{5/8} M_8^{1/8} \dot{m}_{-2}^{-5/8} \beta_{0.5}^{-1/8} \eta_{tur,1.5}^{-3/4} \, \text{MeV}, \quad (16)$$

where we use $q = 5/3$ for the last equation. The turbulence power below the mean thermal proton energy, $3k_B T_p \sim 35\mathcal{R}^{-1}$ MeV, is significantly reduced due to dissipation by plasma kinetic effects, and the turbulent re-energization is not expected when $\varepsilon_{e,crit} < 3k_B T_p$. In our model A (fiducial parameters), we cannot expect re-energization of secondary pairs even for the case with $L_{H\alpha} = L_{min}$. For a further lower $\dot{m}$ case, re-energization may occur. In such a case, we can ignore the inverse Compton component by the secondary pairs owing to its lower photon energy density ($B^2 \propto \dot{m}$, $L_{bol} \propto \dot{m}^2$). Then, the synchrotron peak energy for the re-energized pairs are estimated to be

$$\varepsilon_{\gamma,syn} \approx \frac{3h_p \varepsilon_{e,crit}^2 eB}{4\pi m_e^3 c^5} \simeq 0.07 \left( \frac{\varepsilon_{e,crit}}{100 \text{MeV}} \right)^2 B_2 \, \text{eV}. \quad (17)$$

Hence, we conclude that the re-energized pairs cannot contribute to the MeV gamma-ray background for all the $\dot{m}$ range in our model. Also, primary electrons are not expected to be produced efficiently in RIAFs, because of their rapid thermalization in the range of our interest[28]. Thus, they do not contribute to the MeV gamma-ray background.

**Cumulative background intensities.** Here we describe the method to obtain the background intensities. Since the H$\alpha$ luminosity functions include much fainter sources than the X-ray luminosity functions, we use the luminosity function for type-1 Seyfert galaxies provided by Ref. [45]: $d\rho/dL_{H\alpha} \approx (\rho_*/L_*)/[(L_{H\alpha}/L_*)^{s_1} + (L_{H\alpha}/L_*)^{s_2}]$, where $\rho_* \approx 1.2 \times 10^{-6} \text{Mpc}^{-3}$, $L_* = 3.7 \times 10^{42} \text{erg s}^{-1}$, $s_1 = 2.05$, and $s_2 = 5.12$ (the values of $L_*$ and $\rho_*$ are corrected using Hubble constant of $H_0 = 70$ km s$^{-1}$Mpc$^{-1}$). Type-1 Seyfert galaxies exhibit broad emission lines that are unique to AGN. We extrapolate this luminosity function to $L_{min} = 10^{38}$ erg s$^{-1}$, below which the Palomar survey finds a hint of a break[26]. The survey also indicates a correlation between X-ray luminosity, $L_X$, and $L_{H\alpha}$ for LLAGN. The ratio, $\kappa_{X/H\alpha} = L_X/L_{H\alpha}$ ranges $5 \lesssim \kappa_{X/H\alpha} \lesssim 7$ in the luminosity range of our interest for type-1 AGN. We use $\kappa_{X/H\alpha} = 6.0$ for simplicity[26], but the difference from the cases with $\kappa_{X/H\alpha} = 5$ or $\kappa_{X/H\alpha} = 7$ is less than a factor of 1.2.

Observationally, the X-ray luminosity at the 2–10 keV band can be converted to the bolometric luminosity using the bolometric correction factor, $\kappa_{bol/X} = L_{bol}/L_X \simeq 15$ for LLAGN[97-100]. Using the two correction factor, $\kappa_{bol/X}$ and $\kappa_{X/H\alpha}$, we can convert $\dot{m}$ to $L_{H\alpha}$ if we fix a SMBH mass, $M$. Ref. [101] provided a sample of LLAGN, and the mean and median values of $\log(M/M_\odot)$ are 8.0 and 8.1, respectively. Also, the X-ray luminosity density is dominated by AGN with $M \sim 10^8 - 3 \times 10^8 M_\odot$ if the Eddington ratio function is independent of the SMBH mass[97,102,103]. Thus, we

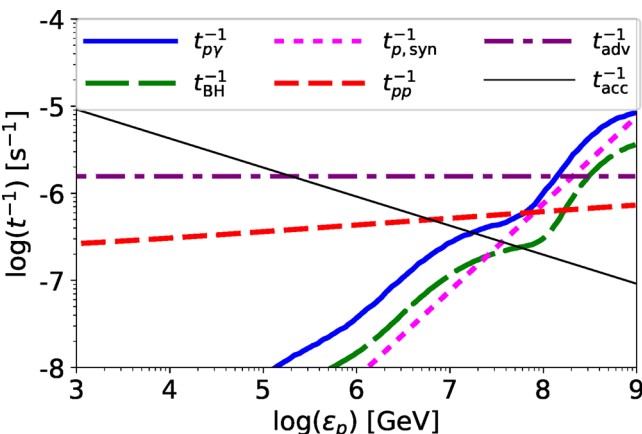

**Fig. 7 Acceleration and energy loss rates as a function of the proton energy for NGC 4579.** The thick-solid, thick-long-dashed, thick-dotted, thick-short-dashed, and thick-dotted-dashed lines are energy loss rates by photomeson production, Bethe-Heitler, synchrotron, $pp$ inelastic collision, and infall processes, respectively. The thin-dotted line indicates the acceleration rate. We use $M = 7.2 \times 10^7 M_\odot$ and $\dot{m} = 8.0 \times 10^{-3}$ with a parameter set for model A (reference model: see Table 1).

set $M = 10^8 M_\odot$ as a reference value. With this prescription, the critical H$\alpha$ luminosity above which RIAFs no longer exist is estimated to be $L_{crit} = \eta_{rad,sd} \dot{m}_{crit} L_{Edd} / (\kappa_{X/H\alpha} \kappa_{bol/X}) \simeq 4.2 \times 10^{41}$ erg s$^{-1}$. Finally, we integrate the gamma-ray and neutrino fluxes over the range of $L_{min} \leq L_{H\alpha} \leq L_{crit}$, which corresponds to $4.6 \times 10^{-4} < \dot{m} < 0.03$ for $\alpha = 0.1$. The background intensity is written as[104]

$$E_i^2 \Phi_i = \frac{c}{4\pi H_0} \int dz \frac{1}{\sqrt{(1+z)^3 \Omega_m + \Omega_\Lambda}} \int dL_{H\alpha} \frac{d\rho}{dL_{H\alpha}} E_i L_{E_i}, \quad (18)$$

where we use the cosmological parameters of $\Omega_m \approx 0.3$ and $\Omega_\Lambda \approx 0.7$. Since dimmer AGN tend to have weaker redshift evolution based on the X-ray, gamma-ray, and radio luminosity functions[103,105,106], no redshift evolution of the H$\alpha$ luminosity function is used. With this treatment, LLAGN at $z < 0.5$ contribute about a half of the diffuse intensity, objects at $0.5 < z < 2$ provide the other half, and the contribution from $z > 2$ is negligible. We consider the gamma-ray attenuation by the extragalactic background light, and include the exponential suppression with the optical depth given in Ref.[107]. The two-photon pair annihilation initiates intergalactic electromagnetic cascades[108], but it has little influence on the GeV gamma-ray intensity because TeV gamma-rays cannot escape from the RIAFs due to two-photon annihilation inside the RIAFs (see Table 2 for the value of $\varepsilon_{\gamma\gamma}$, the $\gamma\gamma$ break energy above which two-photon annihilation attenuates the gamma-rays inside the RIAFs).

Ref.[45] also provided the luminosity function for type-2 AGN that exhibit only a narrow emission line feature, but we find that their contributions to the MeV gamma-ray and PeV neutrino backgrounds are negligible because of two reasons. One is the value of $\kappa_{X/H\alpha}$. Type-2 AGN have $\kappa_{X/H\alpha} \sim 1$[26], and neutrino and gamma-ray luminosity scales with $L_\nu \propto \dot{m}^2 \propto L_{bol} \propto \kappa_{X/H\alpha}$. The other is the shape of the luminosity function. The break luminosity for type-2 AGN is lower than the critical luminosity, $L_* \approx 2.8 \times 10^{40}$ erg s$^{-1} < L_{crit}$, resulting in a lower MeV gamma-ray intensity. The slope in the low-luminosity range is lower than 2, $s_1 = 1.77$, which makes the contribution by relatively faint LLAGN smaller, leading to a lower PeV neutrino intensity. In addition, type-2 objects may be contaminated by non-AGN objects, the fraction of which is still uncertain. Therefore, we focus on contribution by type-1 AGN in this study.

Note that the H$\alpha$ luminosity function does not match the X-ray luminosity function with a constant $\kappa_{X/H\alpha}$. The conversion factor depends on the luminosity, and there is some degree of uncertainty in the conversion factor (see Ref.[52]). The H$\alpha$ luminosity function by Ref.[52] matches the observed X-ray luminosity function at a high luminosity range, while it underestimates the number density for $L_X \lesssim 10^{42}$ erg s$^{-1}$. This range is the most relevant to the MeV gamma-ray and PeV neutrino backgrounds, and the luminosity function by Ref.[45] gives a higher number density at the range. From this reason, we use one by Ref.[45] as a reference case. Future optical spectroscopic and hard X-ray surveys are necessary to reduce this uncertainty.

**Neutrino detectability from nearby LLAGN**. The number of through-going muon track events is calculated by[50,109]

$$\mathcal{N}_\mu(>E_\mu) = \int_{E_\mu}^\infty dE'_\mu \frac{\mathcal{N}_A \mathcal{A}_{det}}{\alpha_\mu + \beta_\mu E'_\mu} \int_{E'_\mu}^\infty dE_\nu \phi_\nu \sigma_{CC} \exp(-\tau_{\nu N}), \quad (19)$$

where $E_\nu$ is the incoming neutrino energy, $E_\mu$ is the muon energy, $\phi_\nu = \Delta t L_{E_\nu} / (4\pi d_L^2 E_\nu)$ is the neutrino fluence from a LLAGN for a time interval of $\Delta t$, $\mathcal{N}_A$ is the Avogadro Number, $\sigma_{CC}$ is the charged-current cross section, $\tau_{\nu N}$ is the optical depth for the neutrino scattering in the Earth, and $\alpha_\mu + \beta_\mu E_\mu$ represents the energy loss rate of muons. We use $\Delta t = 10$ yr and the list of LLAGN given by Ref.[101] (see Ref.[29] for the brightest 10 LLAGN in the list). This method can reproduce the effective area by Ref.[110]. For IceCube-Gen2, we use a $10^{2/3}$ times bigger $A_{det}$ than that for IceCube[111]. The main background of the astrophysical neutrino is atmospheric background, whose intensity depends on the declination. We appropriately take into account both conventional and prompt atmospheric muon neutrinos as well as the attenuation in the Earth[50].

**Power-law injection models for CRs**. The CR spectrum inside the RIAFs can be obtained by solving the transport equation with a power-law injection term:

$$\frac{d}{d\varepsilon_p}\left(-\frac{\varepsilon_p}{t_{cool}} N_{\varepsilon_p}\right) = \dot{N}_{\varepsilon_p, inj} - \frac{N_{\varepsilon_p}}{t_{esc}}, \quad (20)$$

$$\dot{N}_{\varepsilon_p, inj} = \dot{N}_0 \left(\frac{\varepsilon_p}{\varepsilon_{p,cut}}\right)^{-s_{inj}} \exp\left(-\frac{\varepsilon_p}{\varepsilon_{p,cut}}\right), \quad (21)$$

where $\varepsilon_{p,cut}$ is the cutoff energy for the injected protons and $\dot{N}_0$ is the normalization factor. The injection is normalized such that $\int \varepsilon_p \dot{N}_{\varepsilon_p, inj} d\varepsilon_p = \eta_{CR} \dot{m} L_{Edd}$ is satisfied. We estimate $\varepsilon_{p,max}$ by equating the infall timescale to the acceleration timescale, $t_{acc}$. We set $t_{acc} = \eta_{acc} r_L / c$, where $\eta_{acc}$ is the acceleration efficiency parameter. The value of $\eta_{acc}$ is quite uncertain, but $\eta_{acc} \gg 1$ is expected in the RIAFs because

$\eta_{acc} \sim 1$ is achieved only for relativistic shocks or relativistic reconnections (i.e., magnetic energy density is higher than the rest mass energy). Here, we tune it so that our RIAF models can explain IceCube neutrino data. Equation (20) has an analytic solution:

$$N_{\varepsilon_p} = \frac{t_{cool}}{\varepsilon_p} \int_{\varepsilon_p}^\infty d\varepsilon'_p \dot{N}_{\varepsilon'_p, inj} \exp\left(-\mathcal{G}(\varepsilon_p, \varepsilon'_p)\right), \quad (22)$$

$$\mathcal{G}(\varepsilon_1, \varepsilon_2) = \int_{\varepsilon_1}^{\varepsilon_2} \frac{t_{cool}}{t_{esc}} \frac{d\varepsilon'_p}{\varepsilon'_p}. \quad (23)$$

The cooling and loss processes are the same with the stochastic acceleration model. See Ref.[29] for details.

## Data availability
The data generated in this study have been deposited in https://doi.org/10.6084/m9.figshare.15170790. The observational data are obtained from the tables and fitting results provided in the references shown in Figure captions. The extracted data files are available upon reasonable request.

## Code availability
The codes used for this study are available from S.S.K. and K.M. upon reasonable request.

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

## Acknowledgements

This work is supported in part by the IGC postdoctoral fellowship program, JSPS Oversea Research Fellowship, JSPS Research Fellowship, KAKENHI No. 19J00198 (S.S.K.), the Alfred P. Sloan Foundation, NSF Grant No. AST-1908689, and KAKENHI No. 20H01901 and No. 20H05852 (K.M.), and the Eberly Foundation (P.M.).

## Author contributions

K.M. provided the basic idea of this study, and S.S.K. constructed the model and performed all the calculations. K.M. developed the calculation code for the proton-induced cascade emission. All the authors (S.S.K., K.M., and P.M.) discussed the implications of the results, and contributed to the paper.

## Competing interests

The authors declare no competing interests.
