## [Peer Review File · Nature Communications]

Editorial Note:

This manuscript has been previously reviewed at another journal that is not operating a transparent peer review scheme. This document only contains reviewer comments and rebuttal letters for versions considered at *Nature Communications*.

Parts of this peer review file have been redacted as indicated to maintain the confidentiality of other journals.

REVIEWER COMMENTS

Reviewer #1 (Remarks to the Author):

The authors have responded to my remaining comments to my satisfaction. I think the manuscript is ready to be published.

Reviewer #2 (Remarks to the Author):

The authors of the manuscript "Soft gamma-rays from mellow supermassive black holes and connection to energetic neutrinos" propose LLAGN as the source population that could explain both, the MeV-gamma-ray as well as PeV-neutrino background at the same time.

The original manuscript (previously submitted to [redacted]) has been further revised and modified (including the title) with the aim for publishing in Nature Communications now. Indeed, I find the clarity of this manuscript has improved meanwhile significantly.

The authors have also addressed all my comments to the previous manuscript to my satisfaction. Overall, I find this work timely, and potentially suitable for publication in Nature Communications. I have no further significant physics concerns, but only a few minor items for consideration:

- 1) The authors propose - supported by good arguments - that LLAGN could potentially account for the MeV-background, but definitive proof for this scenario can only come from future MeV instruments (such as AMEGO, etc). The second sentence in the abstract should be therefore formulated accordingly, that "LLAGN naturally could account for the MeV gamma-ray background".

2) Not only radio-loud AGN, but also a contribution from star-forming galaxies is necessary/adds to the GeV-gamma-ray background (e.g., Ajello et al 2015, ApJ, 800). This may be added in line 30 of the manuscript.

3) Line 72: Please define the quantity x_M .

4) Eq. 6 uses the scale height H which is, however, defined not before line 226. I suggest to define H around Eq. 6.

Reviewer #3 (Remarks to the Author):

The manuscript discusses a broad range of processes taking place in hot accretion flows. The main focus is given to the Comptonization of the thermal synchrotron emission and acceleration of UHE protons. Both these processes have been already discussed in a number of papers on that and related topics. This poses the first question: it is not clear what is "our model" in this manuscript. It seems that there are only standard ingredients, and the referee could not find any striking interplay of these standard elements. Given that the manuscript is aimed at Nature Communications, this aspect should be presented in a way clear to a broad audience. A related point is that the text is not perfectly accurate. For example, the synchrotron characteristic frequency is dubbed characteristic energy, the Planck constant is not introduced, in lines 96-97 the temperature is discussed in K and mc^2 units (the conversion might be not straight for some readers).

To the referee's understanding, the uncertainties between the processes regulating the MeV emission and neutrino emission are very significant, and the arguments linking these two parts of the paper are vague and not really convincing.

It is unclear why authors in addition to MeV gamma-ray and TeV neutrino also add a short discussion of the EHT observations of M87.

Regarding, the UHE protons, in the manuscript it is stated that the proton maximum energy is 10-100 PeV limited by photohadronic interaction. However, in Fig.5 of methods, it can be seen that t_{fall} is the limiting factor at 1PeV. In the methods, one puts some arguments to explain why this limitation is not rigid, but in the first place, one should explain why the $1/t_{\text{fall}}$ term is an accurate way to describe the acceleration to a black hole.

To summarize, the listed above points do not allow the referee to recommend the manuscript for publication in Nature Communications. If authors can strengthen the connection between the two parts of the manuscript, the referee will be happy to change the opinion.

Reviewer #1 (Remarks to the Author):

The authors have responded to my remaining comments to my satisfaction. I think the manuscript is ready to be published.

Authors:

We thank the reviewer again for carefully reviewing our manuscript and providing constructive/critical comments.

Sincerely,

Shigeo S. KIMURA, Kohta MURASE, and Peter MESZAROS

Reviewer #2 (Remarks to the Author):

The authors of the manuscript "Soft gamma-rays from mellow supermassive black holes and connection to energetic neutrinos" propose LLAGN as the source population that could explain both, the MeV-gamma-ray as well as PeV-neutrino background at the same time.

The original manuscript (previously submitted to [redacted]) has been further revised and modified (including the title) with the aim for publishing in Nature Communications now. Indeed, I find the clarity of this manuscript has improved meanwhile significantly.

The authors have also addressed all my comments to the previous manuscript to my satisfaction. Overall, I find this work timely, and potentially suitable for publication in Nature Communications. I have no further significant physics concerns, but only a few minor items for consideration:

- 1) The authors propose - supported by good arguments - that LLAGN could potentially account for the MeV-background, but definitive proof for this scenario can only come from future MeV instruments (such as AMEGO, etc). The second sentence in the abstract should be therefore formulated accordingly, that "LLAGN naturally could account for the MeV gamma-ray background".
- 2) Not only radio-loud AGN, but also a contribution from star-forming galaxies is necessary/adds to the GeV-gamma-ray background (e.g., Ajello et al 2015, ApJ, 800). This may be added in line 30 of the manuscript.
- 3) Line 72: Please define the quantity x_M .
- 4) Eq. 6 uses the scale height H which is, however, defined not before line 226. I suggest to define H around Eq. 6.

Authors:

We have made all the changes mentioned above by the referee accordingly. We thank the reviewer again for carefully reviewing our manuscript and providing constructive/critical comments.

Sincerely,

Shigeo S. KIMURA, Kohta MURASE, and Peter MESZAROS

Reviewer #3 (Remarks to the Author):

The manuscript discusses a broad range of processes taking place in hot accretion flows. The main focus is given to the Comptonization of the thermal synchrotron emission and acceleration of UHE protons. Both these processes have been already discussed in a number of papers on that and related topics.

Authors:

We thank the reviewer for carefully reviewing our manuscript and providing constructive/critical comments. Below, we provides point-by-point replies to the individual comments.

Reviewer #3:

This poses the first question: it is not clear what is "our model" in this manuscript. It seems that there are only standard ingredients, and the referee could not find any striking interplay of these standard elements. Given that the manuscript is aimed at Nature Communications, this aspect should be presented in a way clear to a broad audience.

Authors:

First of all, we would like to stress that we propose RIAFs in LLAGNs as the origin of the cosmic MeV gamma-ray background for the first time. Our RIAF model can reproduce the MeV gamma-ray background without relying on the non-thermal particles. This result changes our perspective to the high-energy Universe, because the transition from thermal to non-thermal Universe is higher than previously expected. In addition, we propose that energetic neutrinos produced in the RIAFs can account for the cosmic PeV neutrino background detected by IceCube. The two high-energy messengers are connected in the same emission sites, RIAFs in LLAGNs. This connection, namely emissions by thermal electrons and non-thermal protons, is naturally expected, because the plasma conditions at RIAFs in LLAGN allow non-thermal proton acceleration, while the electrons there are easily thermalized due to their shorter thermalization timescale. We believe that these results have a significant impact on a broad range of researchers, such as particle physicists, astrophysicists, and plasma physicists. We have clarified these points in Abstract, Introduction, and Discussion sections.

Reviewer #3:

A related point is that the text is not perfectly accurate. For example, the synchrotron characteristic frequency is dubbed characteristic energy, the Planck constant is not introduced, in lines 96-97 the temperature is discussed in K and mc^2 units (the conversion might be not straight for some readers).

Authors:

We have fixed the typo in the equation for synchrotron characteristic energy and used m_{ec}^2 for the unit of temperature.

Reviewer #3:

To the referee's understanding, the uncertainties between the processes regulating the MeV emission and neutrino emission are very significant, and the arguments linking these two parts of the paper are vague and not really convincing.

Authors:

We admit that the uncertainty of the luminosity function is large. However, as we wrote in the Result section, the uncertainty in the RIAF parameters does not affect our main result, because the MeV gamma-ray background intensity depends very weakly on the RIAF parameters. The PeV neutrino background intensity may change if we use different parameters. However, we can calibrate the parameters using the 10-TeV neutrino data that is reproduced by coronae in Seyfert galaxies. Thus, we do not have a large freedom in our parameter choice. We have clarified these points in the Results section.

Reviewer #3:

It is unclear why authors in addition to MeV gamma-ray and TeV neutrino also add a short discussion of the EHT observations of M87.

Authors:

Since the EHT observations gathered a huge attention from a broad audience, we have added the discussion on it. Our model prediction is consistent with the EHT result, which strengthens the model feasibility. Since this is not the key results of our study, we do not mention it in Abstract and the Discussion section.

Reviewer #3:

Regarding, the UHE protons, in the manuscript it is stated that the proton maximum energy is 10-100 PeV limited by photohadronic interaction. However, in Fig.5 of methods, it can be seen that t_{fall} is the limiting factor at 1PeV. In the methods, one puts some arguments to explain why this limitation is not rigid, but in the first place, one should explain why the $1/t_{\text{fall}}$ term is an accurate way to describe the accretion to a black hole.

Authors:

Since the CR component is coupled with the thermal component through the magnetic fields, the bulk velocity of CRs should be the same as the thermal component. We have clarified it in the Method section. The infall timescale into a black hole is given by $t_{\text{fall}} = \int dR/V_R$. With our one-zone approximation, we can write $t_{\text{fall}} \sim R/V_R$. Then, the loss term by the advective escape process is given by $N_{\text{E}_p} / t_{\text{fall}}$. Since we cited appropriate references for details, we think that equations are not necessary here.

Regarding the cutoff energy appearing in the spectrum, the infall loss leads to a very gradual cutoff (see Becker et al. 2006, ApJ; Kimura et al. 2015, ApJ). The cutoff energy in the spectrum is determined by photomeson production, which results in $E_{\text{cut}} \sim 10-100$ PeV for protons in typical RIAFs in LLAGN. Since this is a technical detail and discussed in the references cited in the Method section, we think that the current description is sufficient.

Reviewer #3:

To summarize, the listed above points do not allow the referee to recommend the manuscript for publication in Nature Communications. If authors can strengthen the connection between the two parts of the manuscript, the referee will be happy to change the opinion.

Authors:

The two high-energy messengers, namely MeV gamma-rays and PeV neutrinos, are connected at the same emission site, RIAFs in LLAGNs. The combination of non-thermal protons and thermal electrons is highly expected due to plasma condition there. We have addressed and clarified the issues raised by the reviewer, and have revised the manuscript accordingly. In addition, we have made additional modifications based on the other reviewers' comments, improving the manuscript. The changes are highlighted by red colors. We hope that our revised manuscript is now acceptable for publication in Nature Communications. Thank you again for carefully reviewing our manuscript and providing constructive/critical comments.

Sincerely,

Shigeo S. KIMURA, Kohta MURASE, and Peter MESZAROS

REVIEWER COMMENTS

Reviewer #3 (Remarks to the Author):

The authors develop an idea that hot accretion flows in low-luminosity AGNs are important contributors to the cosmic background fields. More specifically they argue that MeV gamma-ray background field is generated by Comptonization of thermal electrons and non-thermal protons accelerated in the very same environment are responsible for the TeV neutrino background. While it is believed that sub-MeV background can be produced in AGN coronas, the authors suggest that thermal Compton from hot accretion flow can extend somewhat beyond this component. I think the author fails to show in a convincing way that their consideration is able to accurately describe the spectrum close to its turnover. In this domain, there are a few potentially important effects, which are completely ignored in the manuscript. Since one considers thermal mildly relativistic regime, there should be electron-positron-proton plasma. If a significant amount of positrons is present, this can alter Comptonization process. In addition the electron recoil might be important for Comptonization in the high-energy regime. For example, in Pozdnyakov, L.~A., Sobol, I.~M., & Syunyaev, R.~A. \ 1976, Soviet Astronomy Letters, 2, 55 Fig.2b one shows that recoil is important for $h\nu > mc^2$, i.e., this effect needs to be accounted when one discussed MeV band. To illustrate the feasibility of the scenario authors use the approximate method developed in Ref.17 by an overlapping group of authors, however, it is not clear why this approximation should provide an accurate result in the considered range. A small change of the process close to the spectral turnover can dramatically change the expected flux. I'm afraid this point needs to be clarified before I can recommend the manuscript for publication.

Reviewer #3 (Remarks to the Author):

The authors develop an idea that hot accretion flows in low-luminosity AGNs are important contributors to the cosmic background fields. More specifically they argue that MeV gamma-ray background field is generated by Comptonization of thermal electrons and non-thermal protons accelerated in the very same environment are responsible for the TeV neutrino background. While it is believed that sub-MeV background can be produced in AGN coronas, the authors suggest that thermal Compton from hot accretion flow can extend somewhat beyond this component.

Authors:

We thank the reviewer for carefully reviewing our manuscript. Below, we write replies to his/her individual comments.

Reviewer #3:

I think the author fails to show in a convincing way that their consideration is able to accurately describe the spectrum close to its turnover. In this domain, there are a few potentially important effects, which are completely ignored in the manuscript. Since one considers thermal mildly relativistic regime, there should be electron-positron-proton plasma. If a significant amount of positrons is present, this can alter Comptonization process.

Authors:

In our scenario, the photon luminosity by Comptonization is $L_{\gamma} \sim 10^{42}$ erg/s for relatively luminous LLAGN (see Eq. (3)). This leads to the photon number density of $n_{\gamma} \sim L_{\gamma} / (2 \pi R^2 m_{ec}^3) \sim 3.2 \times 10^7 (L_{\gamma}/10^{42} \text{ erg/s}) (R/10^{14.5} \text{ cm})^{-2} \text{ cm}^{-3}$. Then, the optical depth for gamma-gamma pair production is $\tau_{\gamma\gamma} \sim 0.2 \sigma_T n_{\gamma} R \sim 1.4 \times 10^{-3} (L_{\gamma}/10^{42} \text{ erg/s}) (R/10^{14.5} \text{ cm})^{-1}$. Equating the production rate and escape rate of the electron-positron pairs, we have estimated the number density of the pairs, and we have found that it is a few orders of magnitude lower than the number density of the electrons accompanied with protons. Therefore, we can neglect the Comptonization emission by the electron-positron pairs. We have clarified this in Methods.

Reviewer #3:

In addition the electron recoil might be important for Comptonization in the high-energy regime. For example, in Pozdnyakov, L.~A., Sobol, I.~M., & Syunyaev, R.~A. 1976, Soviet Astronomy Letters, 2, 55 Fig.2b one shows that recoil is important for $h\nu > mc^2$, i.e., this effect needs to be accounted when one discussed MeV band. To illustrate the feasibility of the scenario authors use the approximate method developed in Ref.17 by an overlapping group of authors, however, it is not clear why this approximation should provide an accurate result in the considered range. A small change of the process close to the spectral turnover can dramatically change the expected flux. I'm afraid this point

needs to be clarified before I can recommend the manuscript for publication.

Authors:

We agree that the Comptonization spectra is modified by the electron recoil effect. We have implemented the corrected delta function method given in Coppi & Blandford 1990, which approximately takes the recoil effect into account. We have found that the resulting photon cutoff energy is lowered by a factor of 2, compared to the previous results. Nonetheless, even if we take this into account, our RIAF model can reproduce the MeV gamma-ray background for $E_{\text{gamma}} < \sim 3 \text{ MeV}$, and our conclusions are unchanged by this effect. We have made all the calculations again with the new scheme and updated all the figures and tables with a new parameter set. The calculation method is clarified in Methods. We also slightly modified the discussion regarding the neutrino detectability given in Kimura et al. 2019 PRD, because we have changed the parameter set.

We hope that the revised manuscript is now acceptable for publication in Nature Communications. We thank the reviewer again for carefully reviewing our manuscript and providing constructive/critical comments.

Sincerely,

Shigeo S. KIMURA, Kohta MURASE, and Peter MESZAROS

REVIEWER COMMENTS

Reviewer #3 (Remarks to the Author):

First of all, I apologize for delaying the report.

The authors modified the manuscript to account for my criticism from the previous report. Most notably they adopted the approximation from Coppi&Blandford paper to account for the recoil effect. According to their check, the cutoff indeed shifted to lower energies due to this effect. I appreciate very much the effort in adopting a new approach for their calculations. However, I still see a significant shortcoming in the simulations. The adopted method is based on a delta-functional approximation. In general, this type of approximations reproduces reasonably well the main part of spectra, but (due to the obvious reasons) fails close to the cutoff regions. Moreover, if we look into the original paper by Coppi&Blandford, in their Fig.2, we can see that this approximation provides not a perfect result even in a graph where the energy axis covers 8 orders of magnitude. It is not clear either from the paper or from general limitations of delta-functional approximations why this approach could be adopted for a fine diagnostic of the spectrum close to the cutoff region.

Regarding another question from my previous report, the authors provided an estimate on the injection rate of gamma-gamma produced pair (which appeared to be negligible). This however does answer my question, which was more about their assumption regarding the plasma in their model. When plasma gets heated up above the electron rest energy a certain population of electron-positron pair can be created (independently of the pair creation rate). I, therefore, don't find the provided estimate to be an answer to my question.

Thus, the queries from my previous report remain and I unfortunately cannot recommend the manuscript for publication.

Reviewer #4 (Remarks to the Author):

I have gone through the revised paper, the referee's previous report, the authors' response, and the referee's latest report.

The referee is clearly an expert in Comptonization theory, and also to some degree in pair plasmas, and has thus focused on those issues. The dispute appears to involve a minor numerical factor. I thus recommend publishing, but would like to suggest the following:

1. The authors state that they "utilize the modified delta function method" in Coppi & Blandford (1990). Looking at Fig. 2 in C&B90, two different delta-function models are shown, and I cannot tell which of these corresponds to the "modified delta function method" mentioned by the authors. One of the models is quite terrible and misses the truth by an order of magnitude in the region of interest, while the other is perfectly adequate for this application. The authors should clarify which approximation they use, and how well it represents the spectral region beyond the peak.

2. On the pair plasma question, the authors should cite Kusunose & Mineshige (ApJ, 468, 330, 1996). The authors should also mention that pairs can be created by many processes other than gamma-gamma, e.g., e-e, e-p, e-gamma, etc. In the present application the electron number density is greater than the gamma number density, so some of these other channels might be relevant. A sentence or two on this question would be useful.

Reviewer #3:

First of all, I apologize for delaying the report.

The authors modified the manuscript to account for my criticism from the previous report. Most notably they adopted the approximation from Coppi&Blandford paper to account for the recoil effect. According to their check, the cutoff indeed shifted to lower energies due to this effect. I appreciate very much the effort in adopting a new approach for their calculations.

Authors' reply:

We thank the reviewer again for carefully reviewing our manuscript. Below, we provide our replies to the individual comments.

Reviewer #3:

However, I still see a significant shortcoming in the simulations. The adopted method is based on a delta-functional approximation. In general, this type of approximations reproduces reasonably well the main part of spectra, but (due to the obvious reasons) fails close to the cutoff regions. Moreover, if we look into the original paper by Coppi&Blandford, in their Fig.2, we can see that this approximation provides not a perfect result even in a graph where the energy axis covers 8 orders of magnitude. It is not clear either from the paper or from general limitations of delta-functional approximations why this approach could be adopted for a fine diagnostic of the spectrum close to the cutoff region.

Authors' reply:

We appreciate the reviewer for carefully checking our calculation method. We agree that our method is not perfectly precise. However, we believe that our treatment suffices because of two reasons. One is the quality of the MeV gamma-ray data. We use the "corrected" delta-function approximation, where the probability distribution of the scattered photon energy is approximated to the delta function, but the mean energy of the scattered photon is calculated by the exact kernel, i.e., $P(\omega) \propto \delta(\omega - \langle \omega \rangle)$. The error of our treatment is at most 50% based on Fig.2 and the caption of Blandford & Coppi (1990) (compare the lines " $P(\omega) \propto \delta(\omega - \langle \omega \rangle)$ " and "Exact distribution" in Fig. 2 of Coppi & Blandford 1990), whereas the data of MeV gamma-rays typically have systematic errors of a factor 2 or larger. Therefore, we cannot calibrate the model parameters even if we use the more accurate treatment. The other reason is that our treatment provides a conservative result. In Fig. 2 of Coppi & Blandford (1990), we can see that the corrected delta-function method has a steeper cutoff than the exact solution for $\omega > \sim 30$. This means that the exact shape of the Comptonization spectrum can have a shallower spectrum than our results (but within 50%), and thus, the exact method provides somewhat stronger MeV gamma-ray signals. We have clarified these points in Methods Section.

Reviewer #3:

Regarding another question from my previous report, the authors provided an estimate on the injection rate of gamma-gamma produced pair (which appeared to be negligible). This however does answer my question, which was more about their assumption regarding the plasma in their model. When plasma gets heated up above the electron rest energy a certain population of electron-positron pair can be created (independently of the pair creation rate). I, therefore, don't find the provided estimate to be an answer to my question.

Authors' reply:

We agree that the electron-positron pairs can affect the structure of the accretion flows and the resulting emission in some situation, especially if the plasma is in the pair equilibrium. However, as shown below, the pair equilibrium is not expected in RIAFs with our parameter sets, and the electron-positron pairs do not affect the results.

Two main channels of the pair injection are photon-photon and electron-electron interactions (see e.g., Svensson 1982; Kusunose & Mineshige 1996). We have already shown that the photon-photon interactions cannot alter our conclusions. Regarding electron-electron interactions, the pair production rate can be roughly estimated to be $t_{e-e} \sim 1/(n_p \sigma_T \alpha_{em}^2 c)$, where α_{em} is the fine structure constant. In our reference parameter set, we obtain $t_{fall}/t_{e-e} \sim 4 \times 10^{-4}$, and thus, the hot plasma falls to the black hole before producing sufficient amount of electron-positron pairs. Electron-proton, photon-proton, and photon-electron interactions are also negligible, as the pair production rate is at most comparable to that by the electron-electron interactions, as shown in Methods Section. This means that the pair equilibrium is not achieved in RIAFs. Therefore, we can ignore the effects of pair production.

Indeed, Kusunose & Mineshige 1996 calculated the structure of hot accretion flows taking into account all the relevant pair production processes. They demonstrated that the pair does not affect the dynamical structure of the accretion flows as long as the mass accretion rate is lower than the Eddington rate, or equivalently, as long as we use the viscous $\alpha < 1$. Hence, the pairs are irrelevant in RIAFs with our parameter set ($\alpha=0.1$, or $\dot{m}_{max} \sim 0.03$).

We have clarified these points in Methods Section.

Reviewer #3:

Thus, the queries from my previous report remain and I unfortunately cannot recommend the manuscript for publication.

Authors' reply:

As we justified in the replies above, we have tried to address the reviewer's queries/concerns in the revised manuscript. The changes are highlighted by the red color. We hope that the revised manuscript is now acceptable for publication in Nature Communications. Again, we thank the reviewer for carefully reviewing

our manuscript and providing critical comments.

Sincerely,

Shigeo S. KIMURA, Kohta MURASE, and Peter MESZAROS

Reviewer #4:

I have gone through the revised paper, the referee's previous report, the authors' response, and the referee's latest report.

The referee is clearly an expert in Comptonization theory, and also to some degree in pair plasmas, and has thus focused on those issues. The dispute appears to involve a minor numerical factor. I thus recommend publishing, but would like to suggest the following:

Authors' reply:

We thank the reviewer for providing the constructive and positive comments. Below, we provide our replies to the individual comments.

Reviewer #4:

1. The authors state that they "utilize the modified delta function method" in Coppi & Blandford (1990). Looking at Fig. 2 in C&B90, two different delta-function models are shown, and I cannot tell which of these corresponds to the "modified delta function method" mentioned by the authors. One of the models is quite terrible and misses the truth by an order of magnitude in the region of interest, while the other is perfectly adequate for this application. The authors should clarify which approximation they use, and how well it represents the spectral region beyond the peak.

Authors' reply:

Indeed, we used the better one of the two delta-function methods in Coppi & Blandford (1990). The error by this method is less than 50% based on Fig. 2 in Coppi & Blandford 1990. We have clarified this in Methods Section. This method results in a soft cutoff spectrum, compared to the exact solution. Thus, our results of the MeV gamma-rays are conservative. We have added this comment in Methods Section.

Reviewer #4:

2. On the pair plasma question, the authors should cite Kusunose & Mineshige (ApJ, 468, 330, 1996). The authors should also mention that pairs can be created by many processes other than gamma-gamma, e.g., e-e, e-p, e-gamma, etc. In the present application the electron number density is greater than the gamma number density, so some of these other channels might be relevant. A sentence or two on this question would be useful.

Authors' reply:

We thank the reviewer for letting us know the relevant reference. The pair production rate by e-e interactions can be roughly estimated to be $t_{\{e-e\}} \sim 1/(n_p \sigma_T \alpha_{\{em\}}^2 c)$, where

α_{em} is the fine structure constant. In our reference parameter set, we obtain $t_{\text{fall}}/t_{\text{e-e}} \sim 4.2 \times 10^{-4}$, and thus, e-e interactions cannot produce sufficient amount of pairs. The pair production rate by the e-p, e-gamma, and p-gamma interactions are at most comparable to that by e-e interactions. Thus, we can ignore the effects of the pair production. We have clarified this and cited Kusunose & Mineshige 1996 in Methods Section.

We hope that the revised manuscript is now acceptable for publication in Nature Communications. The changes are highlighted as the red color. We thank the reviewer again for carefully reviewing our manuscript and providing constructive and positive comments.

Sincerely,

Shigeo S. KIMURA, Kohta MURASE, and Peter MESZAROS